# Novel Magnetic Composite Materials for Dental Structure Restoration Application

**DOI:** 10.3390/nano13071215

**Published:** 2023-03-29

**Authors:** Izabell Crăciunescu, George Marian Ispas, Alexandra Ciorîța, Cristian Leoștean, Erzsébet Illés, Rodica Paula Turcu

**Affiliations:** 1National Institute for Research and Development of Isotopic and Molecular Technologies, 400293 Cluj-Napoca, Romania; george.ispas@itim-cj.ro (G.M.I.); alexandra.ciorita@itim-cj.ro (A.C.); rodica.turcu@itim-cj.ro (R.P.T.); 2Faculty of Biology and Geology, Babes-Bolyai University, 400084 Cluj-Napoca, Romania; 3Department of Food Engineering, University of Szeged, 6724 Szeged, Hungary; illes.erzsebet@chem.u-szeged.hu

**Keywords:** magnetic composites, nanoparticles, dental application

## Abstract

In general, magnetic nanoparticles are not often used in dental applications due to some limitations of these materials, such as aggregation problems and low mechanical and chemical resistance but also esthetic problems due to their black color. Our research presents the synthesis of novel magnetic dental composite materials based on magnetic nanoparticles, functionalized and properly coated to overcome the limitations of using magnetic nanoparticles in dental applications. The composites were prepared using a preparation flow containing several integrated reaction steps used previously sequentially. An adequate and deep characterization of dental magnetic composites has been carried out in order to demonstrate that each limitation has been successfully overcome. It was proved that each component brings particular benefits in dental interventions: Fe_3_O_4_ nanoparticles have biocompatible, non-toxic properties and also antimicrobial effects; the SiO_2_ layer significantly increases the mechanical strength of the material; and the Ca(OH)_2_ layer initiates local calcification and significantly improves the color of the dental composite material. Due to magnetic properties, an innovative application approach on the tooth surface can be achieved under an external magnetic field, which, compared to conventional methods, has a major impact on reducing the occurrence of dental caries under filling materials as well as on reducing microfractures.

## 1. Introduction

Dental materials are specially developed materials formulated for use in dentistry [1,2]. These materials have to fulfill several physical, chemical and mechanical requirements: biocompatibility, strong and fracture-resistance, low thermal and electrical conductivity, sufficient elasticity, low fragility and transparency, ease of manufacturing and processing, tasteless and odorless properties, ease of maintenance or repair, aesthetics and not least a reasonable cost [3,4].

There are many different types of dental material, and their characteristics vary according to their intended purpose [5]. There is a general classification of dental materials into four main classes: metallic, ceramic, polymeric and composite type. Each of these classes has certain advantages and disadvantages in their use in dental applications. In general, none of the classes of materials possess all the desired properties of dental materials, so it is essential to combine the properties of materials from different classes. For example, using metal-fused porcelain combines the strength and ductility of metal with the aesthetics of dental porcelain. A polymer or ceramic base can be successfully used for the thermal insulation of a metal paste used in dental restoration. A polymeric material with high thermal expansion but low mechanical resistance and elastic modulus can be reinforced with a ceramic material with low thermal expansion but high mechanical resistance and elastic modulus as a composite filling material. The examples listed above support the idea that understanding both the advantages and limitations of different types of materials allows making a clear selection depending on the intended application [6,7,8,9].

As happened in many fields of science, nanotechnology has also been implemented in the dental area as a great potential opportunity for the development of new nanomaterials for dentistry with improved and unique physicochemical characteristics, such as very small sizes, high area to weight ratio, enhanced chemical reactivity and antimicrobial activity [10,11,12,13,14]. The most widely used nanoparticles (NPs) in dental applications are silver and gold nanoparticles due to their antimicrobial properties, but it is known that silver has a damaging effect on living cells if it is present in a sufficient dose [15,16]. Another category of nanoparticles very often used in dental applications are carbon-based nanoparticles (carbon nanotubes, graphene) because of their excellent mechanical and electrical properties such as heat stability, heat transmission efficiency, high strength and lower density [17,18]. Thus, based on these unique electrical and mechanical properties, carbon-based nanomaterials are important candidates especially for teeth filling but also for other dental applications. TiO_2_ and SiO_2_ nanoparticles are also very often used in the field of dentistry because they significantly decrease microbial adhesion for result promotion [19,20].

Regarding magnetic nanoparticles, the most commonly used magnetic materials with applications in nanotechnology have been iron oxides (FeOx), which have been successfully used in biomedical applications due to their biocompatibility properties and lack of toxicity to human cells [21,22,23]. Among iron oxides, the most common magnetic materials used in nanomedicine are magnetite nanoparticles (Fe_3_O_4_), which due to their superparamagnetic properties can be used in in vivo applications [24,25]. In particular, for dental application, magnetic nanoparticles (MNPs) show high potential in microbial infection control applications, as iron oxide nanoparticles are widely used for biofilm eradication in general and biofilm eradication on dental implants in particular [26,27,28].

Over time, extensive and detailed investigations have been carried out to highlight the possible toxic effects of magnetic nanoparticles on animals and human cells, as well as the effects of tissue exposure to the magnetic field. The toxic effects of different types of MNPs have been determined in both in vitro and in vivo experiments, with research going on as far as application in clinical trials. Based on these studies, iron oxide nanoparticles are considered to be biocompatible materials and have been approved by food and drug administrations for clinical application [29].

As an indispensable tool, MNPs play an important role in all those applications because, under an external magnetic field, the MNPs could directly induce cell effects when exposed to an alternating magnetic field (AMF) [30] and are able to be used to kill tumor cells, for example [31,32].

Most studies reported in the literature related to the applications of MNPs in dentistry use this type of nanoparticles for two main purposes: in prevention and treatment of oral infections, mainly biofilm-related disorders, where magnetic nanoparticles-based dental materials show effective results in the regeneration of dental pulp tissue [33,34,35]; and in dental prosthetics, where magnetic composites are used to generate an adequate magnetic field to promote easier bone healing [36,37]. While magnetic materials have gained real interest for the construction of prosthetic and restorative elements in dental applications as well as for biofilm eradication on dental implants, with a multitude of research showing the development of such systems, there are a limited number of publications on dental restorative applications using magnetic nanoparticles as a component of the final composite material. One of the few scientific research publications in this field, Ref. [38] reported the development of a new innovative magnetic dental composite material with self-adhesion properties that reduces the thickness of the adhesive layer in order to reduce the risk of microleakage.

It can be concluded that magnetic nanoparticles are not often used in dental aesthetic or tooth structure restoration applications due to some limitations of these materials. One of the major problems of these materials is aggregation problems due to strong magnetic interactions between nanoparticles whose surface is not coated. Functionalization of the surface with a thin biocompatible layer is one of the best solutions to avoid aggregation and does not significantly decrease the magnetic properties of the composite material. Another important limitation in using magnetic nanoparticles in dental restorative applications is their low mechanical and chemical resistance. Magnetic nanoparticles are neither chemically inert to the action of acids and/or bases nor sufficiently resistant to various possible mechanical procedures. It is necessary to coat the surface of the magnetic nanoparticles with more resistant molecular layers to provide strength of the composite material. However, one of the major limitations in the application of magnetic composites in aesthetic or restorative dental applications is their color. Magnetic nanoparticles have a dark, black color, so their use in combinations of dental materials would dramatically change the color of the dental composite and thus have a negative impact on the final aesthetics of the dental restoration. Thus, it is very important to find an external coating that will bring major improvements in color.

In the particular case of using MNPs for dental application, even though there are not many studies on the effect of MNPs on dental tissues, based on the well-studied and known physicochemical properties of MNPs, we can assume that concerns about the toxic effect of MNPs are minimal. The as-proposed magnetic nanoparticles are functionalized and properly coated to be biocompatible and have non-toxic properties on the human cell; in addition, they are inserted in dental adhesive material that minimizes contact with human tissue.

In terms of exposure of human tissues to the magnetic field, in the case of using MNPs in dental applications, the dental composite material containing the MNPs are applied on the tooth surface using a permanent magnet applicator. In our previous research [38], preliminary studies were carried out on the use of composite materials based on magnetic nanoparticles and their comparative application, both by classical and magnetic methods. The permanent magnetic applicator uses relatively powerful neodymium magnets with maximum magnetic field intensities of 1 T. Neodymium magnets are commonly used in different technological applications, including magnetic separators, filters, ionizers, in production of on–off buttons, safety sector and security systems [39], but also in the health sector where neodymium magnets are incorporated in medical devices, for example in magnetic resonance imaging devices to diagnose and treat chronic pain syndrome, arthritis, wound healing, insomnia, headache and several other diseases due to their ability to generate a static magnetic field [40]. NASA uses neodymium magnets to maintain the muscular tonus of astronauts during space flights [41].

Neodymium magnets have push–pull forces and have been used as a motion-generating device in orthodontic treatments, molar distillation and palatal expansion [42,43].

Static magnetic fields have been reported to stimulate bone formation via osteoblastic differentiation or activation [44,45].

The main purpose of our article was the synthesis of novel magnetic dental composite materials based on magnetic nanoparticles, functionalized and properly coated to overcome the limitations of using magnetic nanoparticles in dental aesthetic or tooth structure restoration applications. Magnetic materials were synthesized in the form of spherical hydrophobic magnetite nanoparticles (Fe_3_O_4_) with an average size of around 10 nm by the thermal decomposition method, transferred into hydrophilic nanoparticles by an oxidative scission reaction and successively coated with a double layer of silicon dioxide (SiO_2)_ and calcium hydroxide (Ca(OH)_2)_ using a preparation flow containing integrated several reaction steps used previously sequentially. An adequate and deep characterization of dental magnetic composites has been carried out, which is of paramount importance for the development of a well-defined formulation with therapeutic relevance for the targeted application.

The novelty of this article regards the fact that the new dental composite material has been designed to exhibit appropriate physicochemical and mechanical properties to be successfully applied in aesthetic restorative applications of dental structure due to the specific components that provide biocompatibility, chemical and mechanical stability but at the same time bring aesthetic improvements in terms of color. The dental composite material has been specifically developed so that each individual component provides particular benefits in specific dental interventions. Thus, because the as-prepared Fe_3_O_4_ nanoparticles have superparamagnetic properties, they are biocompatible, non-toxic on the human cell and present an antimicrobial effect on bactericidal cultures; all these properties make it a suitable candidate for successful use in dental microbial infection control applications, such as direct and easy eradication of bacterial biofilms from dental implants. The silicon dioxide layer, SiO_2_, has been so designed to significantly increase the mechanical strength of the final dental material, bringing also improvements in color. The presence of the calcium hydroxide layer, Ca(OH)_2_, brings improvements in pulpal desensitization but also has the effect of initiating local calcification and stimulating secondary dentin formation. Moreover, the calcium hydroxide layer, Ca(OH)_2_, is radio-opaque and acts as good thermal and electrical insulation, further improving the color of the dental composite material.

Another innovative idea of this article is that, based on the presence of the magnetic phase in composition, application on the tooth surface will be achieved under an external magnetic field, which compared to conventional application methods has a major impact on reducing and homogenizing the thickness of the adhesive layer and reducing the occurrence of dental caries under filling materials as well as on reducing microfractures. A preliminary study that supports this idea is presented in our previous research, in which it is shown that the incorporation of magnetic nanoparticles into dental adhesives can reduce the thickness of the adhesive layer by 30% after applying a magnetic field on the tooth surface for 2 min and by 86.5% after applying the same magnetic field for 5 min, compared to the application of dental adhesives by conventional techniques [46]. In addition, the dental composite material is prepared by a combined preparation method, which involves several simple, cost-effective and environmentally friendly reaction steps, thus giving the final material a relatively low-cost price.

## 2. Materials and Methods

Iron (III) acetylacetonate (Fe(acac)_3_), oleic acid (OA), oleylamine (Oam), sodium metaperiodate (NaIO_4_), dibenzyl ether, ethyl acetate, acetonitrile and hexane were used for nanoparticles synthesis; while tetraethyl orthosilicate (TEOS, 98%), absolute ethanol, ammonia solution (25%) for silica coverage, calcium nitrate tetrahydrate (Ca(NO_3_)_2_·4H_2_O) and sodium hydroxide (NaOH) were applied for calcium-based coverage. All chemicals were purchased from Sigma-Aldrich, Taufkirchen, Germany and used as received.

The morphology of the magnetic nanoparticles (Fe_3_O_4_) was investigated by transmission electron microscopy (TEM) measurements. The TEM measurements were performed using a Hitachi HD-2700 scanning transmission electron microscope (STEM), Hitachi High-Technologies Corporation, Tokyo, Japan. The magnetic properties at room temperature were measured with a vibrating sample magnetometer, Xiamen, China. XPS measurements were made using a SPECS spectrometer dual-anode X-ray source Al/Mg and a PHOIBOS 150 2D CCD hemispherical energy analyzer (SPECS, Germany). The XPS survey spectra were recorded at 30 eV pass energy and 0.5 eV/step. The high-resolution spectra for individual elements were recorded by accumulating 10 scans at 30 eV pass energy and 0.1 eV/step. The samples were dried on an indium foil for measurements. Argon ion bombardment of the sample surface was performed and the spectra before and after were recorded. Data analysis and curve fitting was performed using CasaXPS 2.3.19 software with a Gaussian-Lorentzian product functions and a non-linear Shirley background subtraction. The high-resolution spectra were deconvoluted into the components corresponding to particular bond types. 

The color parameters of the magnetic dental composites were measured using a spectrophotometer (UV-Jasco-Spectrophotometer) (Jasco, Easton, PA, USA) and a Color Analysis Program, registered as one of the UV-options.

Dynamic light scattering (DLS) and Zeta potential measurements were performed using a NanoZS apparatus (Malvern Panalytical Ltd, Malvern, UK) with a He-Ne laser (λ = 633 nm). The stock sample solution was diluted with NaCl electrolyte to achieve 0.1 g/L solid content. The pH of the systems was adjusted in the range of 3–10, measured directly before placing the sample in the measuring cell.

The magnetic dental composite material was tested against *Escherichia coli* (c), *Enterococcus faecalis* (ATCC 29212) (ATCC Manassas, VA, USA), *Pseudomonas aeruginosa* (ATCC 27853) (ATCC Manassas, VA, USA) and *Staphylococcus aureus* (ATCC 25923) (ATCC Manassas, VA, USA) through diffusimetric methods. The materials were first applied after dispersion in water, at a concentration of 30 mg/mL, and loaded on Wattmann filter discs (20 µL/disc) (Merck KGaA, Darmstadt, Germany). Later on, the final products containing SiO_2_ and Ca were tested against *E. coli* and *P. aeruginosa* for biofilm formation. For this, the materials were pressed to form 6 mm discs that were left in contact with bacterial suspension (0.5 McFarland turbidity in 0.85 NaCl solution) for 24 h at 35 °C. The solution and separate discs were incubated for another 24 h on fresh nutrient agar plates.

Thermal conductivity of the as-prepared magnetic dental composite material was measured using a thermal constants analyzer, Hot Disk TPS 2500S (Kagaku, Göteborg, Sweden).

The electrical conductivity of the magnetic dental composite material was measured with the four-point resistivity measurement method using a Fluke 8845A 6.5 Digit Precision Multimeter (Artisan Technology Group, Champaign, IL, USA).

## 3. Results

### 3.1. Synthesis of Hydrophobic MNPs (Fe_3_O_4_-OA)

The magnetic dental composite material was prepared using a preparation flow containing several integrated reaction steps used previously sequentially and so tailored as to obtain the dental magnetic nanocomposites with specific properties for dental applications in general and to overcome the limitations of using magnetic nanoparticles in tooth structure restoration applications in particular. 

Hydrophobic monodispersed magnetic nanoparticles (MNPs) (magnetite—Fe_3_O_4_) with spherical shape and average size around 10 nm were synthesized by thermal decomposition of iron organometallic compounds (Fe(acac)_3_) in high boiling point solvent (benzyl ether) containing stabilizing surfactants (oleic acid, OA and oleyl amine, OAm). The thermal decomposition method was chosen because it is considered one of the most precise synthesis methods of magnetic nanoparticles. Even though this method is not so simple, it is a useful method to manufacture monodisperse magnetic nanoparticles with control over the size and shape of particles, thus avoiding aggregation problems, one of the major limitations of magnetic nanoparticles. The synthesis procedure is presented schematically in Figure 1 (first reaction step), hydrophobic magnetic nanoparticles being noted as Fe_3_O_4_-OA [47].

In a typical experiment for the synthesis of hydrophobic magnetite nanoparticles (MNPs), Fe(acac)_3_ and a 1:1 molar ratio of OA:Oam were dissolved in benzyl ether (20 mL) and magnetically stirred under a flow of argon. The mixture was heated to 150 °C for 60 min under an atmosphere of argon, to 200 °C for another 60 min and to reflux (293 °C) for another 30 min. The black mixture was cooled to room temperature, washed three times with ethanol (40 mL), precipitated and separated magnetically [48]. Magnetic separation is an easy and highly efficient separation process of separating components of mixtures by using a magnet to attract magnetic substances. In our particular case, the process was used for magnetic separation of magnetite nanoparticles by the reaction mixture, also in the washing process of particles. In a typical separation process, the MNPs colloidal solution was placed on a permanent magnet and left for 10 min to separate by migrating MNPs on the magnet. The supernatant was subsequently discharged; the sample was thoroughly washed and subjected to a new magnetic separation process. At the end the clean MNPs were used in subsequent reaction steps, being redispersed in toluene.

### 3.2. Synthesis of Hydrophilic MNPs (Fe_3_O_4_-AZA)

Due to the fact that, by the thermal decomposition method, size- and shape-controlled nanoparticles can be obtained, they are hydrophobic, and thus they cannot be directly used in biomedical applications. A further functionalization of their hydrophobic surface is required in order to transform them into hydrophilic nanoparticles. It is known that, in general, the transformation processes of hydrophobic particles into hydrophilic ones bring significant negative changes in morphological, structural or magnetic properties by adding new molecular or polymeric layers. Therefore, we propose a very efficient and environmentally friendly method to transfer the hydrophobic magnetic nanoparticles in hydrophilic medium, using oxidative cleavage of the double bond of oleic acid, which is the initial surfactant that covers the hydrophobic magnetic nanoparticles [49]. For the transfer of hydrophobic magnetic nanoparticles into hydrophilic nanoparticles, the oxidative scission reaction of the double bound is applied to convert unsaturated fatty acids covering the nanoparticles surfaces to carboxylic and dicarboxylic acid. The reaction/process is presented schematically in Figure 1 (second reaction step), the hydrophilic magnetic nanoparticles covered with azelaic acid being noted as Fe_3_O_4_-AZA [47,49]. This process allows us not only to obtain the hydrophilic and biocompatible magnetic nanoparticles, but also to preserve the morph structural and magnetic properties of the nanoparticles, because it does not imply any coating with additional layers that might induce their aggregation.

A total of 100 mg of hydrophobic magnetic nanoparticles was dispersed in a mixture of 20 mL of ethyl acetate and acetonitrile (1:1 volume ratio). In the next step, sodium periodate aqueous solution (400 mg NaIO_4_ in 15 mL H_2_O) was added to this solution and the mixture was let to react at room temperature for four hours. Finally, a two-phase mixture was formed containing a colorless layer at the top, which was the organic solvent; the toluene and the hydrophilic magnetic nanoparticles were dispersed in aqueous medium at the bottom phase. The solvent was discharged and the magnetic nanoparticles were magnetically separated (as described previously) from the aqueous medium, washed three times with ethanol and distilled water and finally re-dispersed in water.

### 3.3. Synthesis of SiO_2_-Covered Magnetic Composites (Fe_3_O_4_-SiO_2_)

The as-prepared hydrophilic, biocompatible nanoparticles, Fe_3_O_4_-AZA, with good morpho-structural and magnetic properties still have limitations in terms of using them in dental restorative applications because of their low mechanical and chemical resistance. Thus, coating the surface of magnetic nanoparticles with a layer of controlled thickness of SiO_2_, chemically inert to the action of acids and/or bases but sufficiently resistant to various possible mechanical procedures, will bring significant improvements in the chemical and mechanical stability of the final magnetic composites. The synthesis of SiO_2_ layer on the surface of Fe_3_O_4_-AZA was performed using the modified Stoeber method. The hydrolysis of the SiO_2_ precursor, tetraethyl orthosilicate (TEOS), in basic medium is presented schematically in the Figure 1 (third step), the SiO_2_-covered magnetic composites being noted as Fe_3_O_4_-SiO_2_ [50].

In a typical synthesis of Fe_3_O_4_-SiO_2_, 0.4 g (1 wt%) of dry magnetic nanoparticles (Fe_3_O_4_-AZA) was added to a mixture of ethanol (320 mL) and water (80 mL) with a volume ratio of ethanol: water = 4:1, and the colloidal solution was sonicated using an ultrasonic finger for 10 min. To this Fe_3_O_4_-AZA solution, 8 mL of aqueous ammonia solution (25 wt%) and 6 mL TEOES (18 mM) were added under vigorous magnetic stirring (500 rpm). The reaction mixture was kept at room temperature for one hour and then the product was separated by an external magnet (as described previously) and washed several times with distilled water. The final product was collected and dried at 60 °C for 12 h.

### 3.4. Synthesis of Ca(OH)_2_-Covered Magnetic Composites (Fe_3_O_4_-Ca(OH)_2_)

One of the major limitations in the application of magnetic composites in dental restorative and esthetic applications remains their color. Magnetic nanoparticles have a dark, black color, so their use in dental materials would alter the final color of the dental composite, with a negative impact on the aesthetics of the dental work. Thus, it is important to find an external coating that will bring major improvements in color. Calcium-based inorganic compounds are known to have a very intense white color and enhanced covering capacity. We have chosen calcium hydroxide coating, Ca(OH)_2_, as an external coating because, in addition to a major improvement in the color of the final composite material, it also brings improvements in pulpal desensitization since it has an initiating effect on local calcification and stimulates the formation of secondary dentin. At the same time, being radio-opaque, it acts as good thermal and electrical insulation. Preparation of the Ca(OH)_2_ layer on the surface of Fe_3_O_4_-SiO_2_ composites is achieved by direct reduction of a calcium precursor in NaOH solution and is presented schematically in Figure 1 (fourth step). The Ca(OH)_2_-covered magnetic composites are noted as Fe_3_O_4_-Ca(OH)_2_.

In a typical synthesis of Fe_3_O_4_-Ca(OH)_2_, 94.5 g calcium nitrate tetrahydrate, Ca(NO_3_)_2_ × 4H_2_O, was dissolved in distilled water (8.5 wt%) and 1.2 mL colloidal dispersion (corresponding to 1 g of wet sample) of Fe_3_O_4_-SiO_2_ magnetic composites wass added. To the as-prepared colloidal dispersion/sol, a sodium hydroxide (NaOH) solution (16 g, 1.6%) was added drop by drop at an addition rate of 1 mL/min under strong magnetic stirring (1200 rpm) at room temperature. About half an hour after the start of the addition, the reaction medium becomes turbulent and a white precipitate, Ca(OH)_2_, appears. After 3 h of reaction at room temperature, the solution was filtered and the as obtained precipitate was washed 3 times successively with 100 mL water and dried in an oven for 1 h at 600 °C.

## 4. Discussion

The morphology of magnetic dental composite materials (Fe_3_O_4_-OA, Fe_3_O_4_-SiO_2_, and Fe_3_O_4_-Ca(OH)_2_) is shown by TEM images in Figure 1.

In the TEM images of the hydrophobic MNPs (Fe_3_O_4_-OA), Figure 1A, one can observe a round uniform structure of MNPs; they are well dispersed, with an average diameter of 10 nm. In Figure 1B, the MNPs embedded in the SiO_2_ layer (Fe_3_O_4_-SiO_2_) are presented, being observed as slight aggregations of MNPs during the SiO_2_ coating process. In Figure 1C, the MNPs embedded in the double layer of inorganic shell, SiO_2_ and Ca(OH)_2_, are presented. A relatively uniform distribution of MNPs in the inorganic shell can be observed, and also clear delimitation of the two layers is visible. This well-defined morphology of the dental magnetic composites made it possible to calculate the thicknesses of the two inorganic layers that coat the MNPs, so that the average SiO_2_ layer thickness measured from MNPs to first layer was 26.9 nm ± 2.5 nm (*n* = 4 ± S.D) and the average Ca(OH)_2_ layer thickness measured from first layer to second layer was 19.3 nm ± 3.6 nm (*n* = 6 ± S.D). As a result of morphological characterization, based on the MNPs size and the thickness of the two inorganic layers, it can be concluded in general terms that the as-prepared dental magnetic composite is suitable for dental application because, in particular, each of the components of dental magnetic composites (MNPs, SiO_2_ and Ca(OH)_2_ layers) provides the specific properties to which it has been assigned. The SiO_2_ layer is thick enough to ensure the mechanical and chemical resistance of the dental magnetic composites and Ca(OH)_2_ layer thicknesses are adequate to provide a significant improvement in color of the final dental material. However, neither of the two layers is too thick to change significantly the magnetic properties of the dental magnetic composites provided by magnetic nanoparticles, thereby providing the possibility of application of these materials on the tooth surface under an external magnetic field, an innovative and very effective method relative to conventional ones.

In order to be able to use the as-prepared dental magnetic composites in dental restorative applications under an external magnetic field, the minimum requirement is that this type of material has relatively good magnetic properties so that it interacts properly with the field. Magnetic measurements support the idea of applying the as-prepared dental magnetic material on the tooth surface under an external magnetic field, a method with a major impact on reducing the occurrence of dental caries under filling materials as well as on reducing microfractures [38,46]. In Figure 2, the hysteresis loops at room temperature of the MNPs Fe_3_O_4_-SiO_2_ and Ca(OH)_2_ are presented. A saturation magnetization value of 65.3 emu/g was estimated for the initial MNPs (Fe_3_O_4_-OA), which decreased with the coating of the two inorganic layers to 22.3 emu/g for Fe_3_O_4_-SiO_2_ and to 11.4 emu/g for the final dental magnetic composites, Fe_3_O_4_-Ca(OH)_2_. The decrease of the magnetization value is normal, explained by the coating of the magnetic nanoparticles with non-magnetic layers, the saturation magnetization value still remaining high enough for the intended application.

The chemical compositions of the synthesized magnetic dental composite were determined using XPS analysis. The high-resolution XPS spectra of C1s, O1s, Si2p and Fe2p identified in the sample Fe_3_O_4_-SiO_2_ are shown in Figure 3.

The spectra of C, O and Si before sputtering with argon ions are deconvoluted into the components. The best fit for C1s spectrum was obtained with three components assigned to C–C, CH (284.6 eV), C–O (286.6 eV) and O–C=O (289.3 eV) from the coating of nanoparticles with azelaic acid. The O1s spectrum exhibits one component located at 534.6 eV ascribed to SiO_2_.

The spectrum of Si2p containing the doublet Si2p3/2 and Si2p1/2 from Figure 3 confirms the coating of nanoparticles with SiO_2_. Due to the coating of nanoparticles with SiO_2_, the Fe2p spectrum could be observed only after sputtering at 2000 V for 180 min with argon ions (spectrum (b) in Figure 3).

In Figure 4 are shown the XPS spectra of C1s, O1s, Si2p, Ca2p and Fe2p from the sample Fe_3_O_4_-Ca(OH)_2_. The deconvolution of C1s, O1s, Si2p and Ca2p spectra in the Figure 4 provide evidence of the double layer coating (SiO_2_ and Ca(OH)_2_) of nanoparticles stabilized with azelaic acid. The best fit of C1s spectrum was obtain with two components corresponding to C–C, CH (285 eV) and O–C=O (288.8 eV). O1s spectrum contains two components ascribed to Ca–O and Si–O. As in the case of the Fe_3_O_4_-SiO_2_ sample, the Fe 2p spectrum for the sample Fe_3_O_4_-Ca(OH)_2_ could be observed only after sputtering at 1500 V 60 min and 2000 V 180 min due to the nanoparticles coating with SiO_2_ and Ca(OH)_2_ layers.

In order to highlight the possibility of using the as prepared dental magnetic composites in aesthetic dental restorative applications, color measurements of the materials in each step of the preparation were carried out. The CIELAB color system (L*a*b*) shown in the diagram in Figure 5 was used to provide a mathematical description of a color difference perceived by the human eye [51].

The CIELAB, or CIE L*a*b*, color system represents the quantitative relationship of colors on three axes: L* value indicates lightness, and a* and b* are chromaticity coordinates. On the color space diagram, L* is represented on a vertical axis with values from 0 (black) to 100 (white) [52]. The * value indicates red-green component of a color, where +a* (positive) and −a* (negative) indicate red and green values, respectively. The yellow and blue components are represented on the b* axis as +b* (positive) and −b* (negative) values, respectively. The center of the plane is neutral or achromatic. The distance from the central axis represents the chroma (C*), or saturation of the color. The angle on the chromaticity axes represents the hue (h°) [52].

For color measurement, a standard UV–VIS spectrum was registered in the wavelength range of 780 to 380 nm and the lightness index (L*), chromaticity coordinates (a* b*) and color difference (ΔL*, Δa*, Δb*) were calculated using a Color Analysis Program, registered as one of the UV-options. ΔE* value was determined mathematically using Equation (1):(1)$$\Delta E=\sqrt{{\left(\Delta a*\right)}^{2}+{\left(\Delta b*\right)}^{2}+{\left(\Delta L*\right)}^{2}}$$

Table 1 shows the registered values of brightness L*, color coordinates a* and b*, as well as the degree of color difference ΔE, ∆L*, Δa*, Δb* (calculated between the measured samples and the control sample, a dental standard sample [53] for materials obtained in each reaction step of preparation of dental magnetic composites).

Analyzing the color measurement data, one can conclude that the brightness difference ΔL* decreases in absolute value with the surface coverage of magnetic nanoparticles, indicating brightness increased in the sample. The sample becoming lighter in color compared to the almost black color of the original nanoparticles (the color change can be seen in the images attached to each sample in Table 1), reaching a reasonable value of only 5.07 units compared to the control sample. There is also a significant decrease in the color difference ∆E compared to the standard sample, with a 13.95 units difference compared to the control sample. From visual point of view, a major color improvement is observed by coating the magnetic nanoparticles with a double layer of SiO_2_ and Ca(OH)_2_, and it is also observed that the color of the final magnetic dental composite material conforms to the color of standard samples of dental materials. It is noted that in dental restorative applications, the magnetic dental composite will be mixed with other dental structural materials, which will additionally smooth the final color.

The chemical stability of a composite material used in dental applications is particularly important, due to the fact that these materials are in direct contact with the biological salivary environment and are constantly exposed to various pH environments. So, it is especially important to measure their chemical stability at different pHs. DLS measurements were carried out to study the hydrodynamic diameter of the synthesized material under various solution conditions, providing essential information about their aggregation behavior in aqueous media.

Figure 6 shows the hydrodynamic diameters recorded for magnetic composites in each synthesis step under various pH conditions.

It can be seen from Figure 6 that the hydrodynamic diameter of MNPs and inorganic coated magnetic composites increases compared to particle diameter values determined from TEM. It is known that the hydrodynamic diameter is always larger than the particle diameter, because it takes into account the interactions between the surface and the aqueous environment. In this study, the carboxylic groups of the hydrophilic MNPs (Fe_3_O_4_-AZA), the SiO_2_ and the Ca(OH)_2_ layers all are hydrophilic in nature, which can form hydrogen bonds with the water molecules of the medium, so a slight increase of the hydrodynamic diameter compared to the primary particle size obtained by TEM is normal. In our study, a more significant increase in hydrodynamic diameter was observed in all three cases, which can be attributed to the presence of the electric double layer, whose thickness is greatly influenced by the solution conditions (pH, salt concentration, etc.).

However, no significant changes in the hydrodynamic diameter could be observed at different pHs, which suggests a high stability of the material in this pH range. This is a particularly important conclusion, especially since the dental composite material is used in salivary biological media, with additional pH fluctuations in the same range measured by us.

An essential property of materials used in dental restorative applications is their ability to increase the deposition of calcium and phosphate ions on the dental surface. This property can be estimated by measuring the zeta potential value in aqueous medium. It is already known that negative charges are necessary for bioactive behavior [54], e.g., negative surfaces promote better the apatite formation at the surface. Thus, the negative potential values assess the apatite-forming capability of the materials. In Table 2, the measured zeta potential values at pH = 7 are collected.

In Table 2, a small negative ZP value for MNPs that increases in absolute value with the surface coverage by the double inorganic layers can be observed, which may indicate an enhanced bioactive behavior. It can thus be expected that the final magnetic dental composite material will have an effect on initiating local calcification and stimulating secondary dentin formation.

Bactericidal properties of the as-prepared dental magnetic composites were tested against *Escherichia coli* (ATCC 25922), *Enterococcus faecalis* (ATCC 29212), *Pseudomonas aeruginosa* (ATCC 27853) and *Staphylococcus aureus* (ATCC 25923) using diffusimetric methods [55], and the results are presented in Table 3.

Additional tests for biofilm formation were performed against *E. coli* and *P. aeruginosa,* as well. The materials had a slight bacteriostatic effect, due to the fact that after re-incubation after treatment all plates developed uniform biofilms as compared to Fe_3_O_4_-Ca(OH)_2_, which had bacteria on its surface as well as around it (Figure 7). The tested material exhibits a bacteriostatic effect, which indicates that the materials might not favor the development of bacteria on their surface.

Thermal insulation is one of the key properties of materials used in dental restorative applications. The high thermal conductivity of dental material is a disadvantage because if it is near the pulp, it may cause patient discomfort as a result of temperature changes produced by hot or cold foods. The thermal conductivity is the parameter that defines the thermal insulation properties of materials and is defined as the amount of heat passing through a body when the temperature difference is 1 °C. We propose an innovative magnetic dental composite material having a calcium hydroxide external layer, as Ca(OH)_2_ has been used as a lining and sub-lining material for several decades [56], based on its therapeutic effects on the exposed human pulp and ability to promote dentine bridging [57,58]. The presence of the calcium hydroxide layer, Ca(OH)_2_, brings improvements in pulpal desensitization but also has the effect of initiating local calcification and stimulating secondary dentin formation. Moreover, the calcium hydroxide layer, Ca(OH)_2_, is radio-opaque and acts as good thermal and electrical insulation.

The thermal conductivity of the as-prepared magnetic dental composite material was measured, and the results were compared with other restorative materials and human dentin to examine thermal insulation effects. The experimentally measured thermal conductivity for Fe_3_O_4_-Ca(OH)_2_ sample was 0.675 W/mK, very close to the value of human dentin (0.56 W/m/K). As compared with other types of dental materials, the as-prepared magnetic dental composite material has lower values as follows: composite resin (0.77 W/mK), enamel (1.20 W/mK), zinc phosphate cement (1.13 W/mK) and Au-Ag-Pd (38.5 W/mK) [59]. The thermal conductivity of the as-prepared dental composite material was in the same range as that of human dentin and lower than that of other materials used for dental restoration, suggesting that the composite material has a good thermal insulation effect and can be used successfully in dental applications.

Electrical conductivity is not typically thought of as an important property, but we should know which restorations materials are conductive and which are not. Occasionally, a new amalgam filling will hurt when it is touched with a metal fork. Saliva facilitates the flow of electrons between metals, producing an electrical current like a battery, and patients will experience a pain reaction to electrical current in teeth with deep filling and little insulating dentin. Even if the thermal insulation property is not necessarily an essential condition of a dental restorative material from the point of view of the unpleasant symptoms that a conductive material might have in use in dental restorative applications, it is required that the dental material have electrical insulation properties in addition to thermal insulation effects. The experimentally measured resistivity for the Fe_3_O_4_-Ca(OH)_2_ sample was ρ = 130 MΩ/cm, and considering that electrical resistivity is the opposite condition of electrical conductivity, the conductivity was estimated to be 0.0077 µS/cm, a value considered in the area of insulating materials.

## 5. Conclusions

Magnetic dental composite materials based on magnetite nanoparticles coated with double layer of silicon dioxide, SiO_2_ and calcium hydroxide, Ca(OH)_2_ was prepared using a preparation flow containing four integrated reaction steps. 

The morphology of the magnetic nanoparticles demonstrates the formation of a uniform spherical structure of hydrophobic MNPs, well dispersed, with average diameter of 10 nm uniformly covered with a 26.97 nm ± 2.56 nm (*n* = 4 ± S.D)-thickness SiO_2_ layer and 19.38 nm ± 3.64 nm (*n* = 6 ± S.D)-thickness Ca(OH)_2_ layer. A saturation magnetization value of 11.4 emu/g was measured for the final dental magnetic composites Fe_3_O_4_-Ca(OH)_2_, considered a relatively good value for this kind of composite material, magnetic particles coated with a double layer of inorganic non-conductive material. XPS analysis was used to determine the chemical compositions of the synthesized magnetic dental composites and the structure was confirmed by identifying each of the components of dental magnetic composites (MNPs, SiO_2_ and Ca(OH)_2_ layers).

The color measurements data indicate that the brightness difference (ΔL*) decreases with the surface coverage of magnetic nanoparticles, the sample becoming lighter in color compared to the almost black color of the original nanoparticles. There is also a significant decrease in the color difference (∆E) compared to the standard sample, the color of the final magnetic dental composite material conforming to the color of standard samples of dental materials.

From the chemical stability point of view, no structural changes were observed, such as aggregation phenomena as a result of the destabilization of the inorganic layers covering the MNPs under the action of different pHs. A high negative zeta potential value was determined for the magnetic dental composites (−34.4 mV), indicating a very good bioactive behavior. It can thus be stated that that the final magnetic dental composite material has the effect of initiating local calcification and stimulating secondary dentin formation.

Bactericidal properties of the as-prepared dental magnetic composites were tested against *Escherichia coli*, *Enterococcus faecalis*, *Pseudomonas aeruginosa* and *Staphylococcus aureus* and, in addition, tests for biofilm formation were performed against *E. coli* and *P. aeruginosa*. The as-tested material exhibits a bacteriostatic effect, which indicates that the materials might not favor the development of bacteria on their surface.

## Data Availability

The data presented in this study are available on request from the corresponding author.

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
