# Peer review of "Novel Magnetic Composite Materials for Dental Structure Restoration Application"

_nanomaterials, 2023, doi:10.3390/nano13071215_

Round 1
Reviewer 1 Report
Manuscript reports on the manufacture and characterization of Fe3O4 magnetic nanoparticles covered by SiO2 and Ca(OH)2. The method of the preparation of composites is described in detail and produced species are thoroughly tested. Authors did a lot of work. It is argued that the Fe3O4 superparamagnetic nanoparticles have positive effects on the human cell and also antimicrobial effect. Also cited references [19], [20], [21] report on experimental findings being in agreement with authors' assurance. However, the principal issue of the manuscript is the missing direct evidence of such magnetic nanoparticle functioning. There is a long-term disputation in the scientific community on the effect of magnetic field on living organisms. Tests with both positive and negative results appeared. Since the magnetic field interacts only weakly with common organic matter, it is not clear, how could it affect the organisms existing at the room temperature where the thermal fluctuations significantly violate magnetic forces. Existing physical explanation of observed effects proposes the dominant influence of chemical or electrical interaction of nanoparticles with the tissue instead of magnetic one. It is thus necessary, before the manuscript might be positively assessed, that authors critically review their work and improve their argumentation with respect to the importance of magnetic field on the properties and utility of magnetic nanoparticles.
In addition to the general comment above, authors should comment or amend the manuscript according following items.
1. Authors' principal statement in the Introduction, lines 129-134, "An innovative property ... has a major impact in reducing and homogenizing the thickness of the adhesive layer, ..." and repeated in lines 327-329 "Magnetic measurements support the idea of applying the as prepared dental magnetic material on the tooth surface under an external magnetic field, method with a major impact in reducing the occurrence of dental caries under filling materials as well as in reducing microfractures." should be completed by proper reference attesting to this achievement. If this is the result presented for the first time in this paper, it should be stated explicitly summarizing arguments leading to such conclusion.
2. line 239: The method of separation of magnetic nanoparticles should be specified in more detail.
3. line 336: "the saturation magnetization value still remaining high enough for the intended application." Which criterion was used to support this statement? What is the limiting magnetization acceptable for intended application?
Typing errors
line 12: "Our research presents ..."
line 15: "composites were ..."
line 22: "properties, an innovative ..."
line 152" "an XPS"
line 219: "Therefore, ..."
line 385: "was obtained"
Table 1.: Line numbers interfere in the Table. The same problem is seen in Fig. 6 and Table 3.
Author Response
Dear Reviewer
Please find enclosed our revised manuscript, "Novel magnetic composite materials for dental structure restoration application"
Authors: Izabell Craciunescu *, George Marian Ispas, Alexandra Ciorita, Cristian LeoÈ™tean, Erzsébet Illés, Rodica Paula Turcu
Manuscript ID: nanomaterials-2263544
The authors appreciate the efforts of the reviewers, being also grateful for their remarks and constructive suggestions helping us to improve the clarity of the manuscript. Following their observations, we have done efforts to improve the English and the clarity of the manuscript, doing our best to remove possible misunderstandings. The main amendments are highlighted in red in the present manuscript and a point-by-point response to the reviewer comments are presented in the following (answers in italics):
Manuscript reports on the manufacture and characterization of Fe3O4 magnetic nanoparticles covered by SiO2 and Ca(OH)2. The method of the preparation of composites is described in detail and produced species are thoroughly tested. Authors did a lot of work. It is argued that the Fe3O4 superparamagnetic nanoparticles have positive effects on the human cell and also antimicrobial effect. Also cited references [19], [20], [21] report on experimental findings being in agreement with authors' assurance. However, the principal issue of the manuscript is the missing direct evidence of such magnetic nanoparticle functioning. There is a long-term disputation in the scientific community on the effect of magnetic field on living organisms. Tests with both positive and negative results appeared. Since the magnetic field interacts only weakly with common organic matter, it is not clear, how could it affect the organisms existing at the room temperature where the thermal fluctuations significantly violate magnetic forces. Existing physical explanation of observed effects proposes the dominant influence of chemical or electrical interaction of nanoparticles with the tissue instead of magnetic one. It is thus necessary, before the manuscript might be positively assessed, that authors critically review their work and improve their argumentation with respect to the importance of magnetic field on the properties and utility of magnetic nanoparticles.
“The principal issue of the manuscript is the missing direct evidence of such magnetic nanoparticle functioning”
Answer: In our preliminary articles, reported as references [38], [46] in the revised version
[38]. Cristian Zaharia, Virgil-Florin Duma, Cosmin Sinescu, Vlad Socoliuc, Izabell Craciunescu, Rodica Paula Turcu , Catalin Nicolae Marin, Anca Tudor, Mihai Rominu, Meda-Lavinia Negrutiu, Dental Adhesive Interfaces Reinforced with Magnetic Nanoparticles: Evaluation and Modeling with Micro-CT versus Optical Microscopy, Materials, 2020, Volume: 13 Issue: 18 , 3908
[46]. C.Zaharia, C. Sinescu, A.G.Gabor, V.Socoliuc, S. Talpos, T. Hajaj, P. Sfirloaga, R. Oancea, M. Miclau, M.L.Negrutiu, Influences of Polymeric Magnetic Encapsulated Nanoparticles on the Adhesive Layer for Composite Materials Used for Class I Dental Fillings, MATERIALE PLASTICE, 56, 2, 2019 ]
there are already research proving the usefulness of magnetic nanoparticles in dental restoration applications. The conclusion of this studies demonstrate that the use of magnetic nanoparticles after incorporation into dental adhesives, can reduce the thickness of the adhesive layer by 30% by applying a magnetic field on the tooth surface for 2 min and by 86.5 % for applying the same magnetic field for 5 min compared to the application of dental adhesives by conventional techniques.
“There is a long-term disputation in the scientific community on the effect of magnetic field on living organisms. Tests with both positive and negative results appeared. Since the magnetic field interacts only weakly with common organic matter, it is not clear, how could it affect the organisms existing at the room temperature where the thermal fluctuations significantly violate magnetic forces. Existing physical explanation of observed effects proposes the dominant influence of chemical or electrical interaction of nanoparticles with the tissue instead of magnetic one.”
Answer: In order to clarify the concerns about the effect of the magnetic field on human tissue, we have added in the Introduction, a discussion related to the effects of the alternating magnetic field on the human tissues. Additionally we add a clarification regarding the fact that in our application, we use the magnetic field provided by a permanent magnet (Neodymium magnet) with a maximum intensity of 1T, used strictly for high precision localization of composite dental material. References have also been added to support the idea that, the magnetic field provided by permanent magnets, is not only non-toxic to human tissues but is even used in certain treatment methods.
“Authors' principal statement in the Introduction, lines 129-134, "An innovative property ... has a major impact in reducing and homogenizing the thickness of the adhesive layer, ..." and repeated in lines 327-329 "Magnetic measurements support the idea of applying the as prepared dental magnetic material on the tooth surface under an external magnetic field, method with a major impact in reducing the occurrence of dental caries under filling materials as well as in reducing microfractures." should be completed by proper reference attesting to this achievement. If this is the result presented for the first time in this paper, it should be stated explicitly summarizing arguments leading to such conclusion.”
Answer: In the Introduction, after we stated "An innovative property ... has a major impact in reducing and homogenizing the thickness of the adhesive layer, ..." , we have added a text in which we briefly conclude the impact of MNPs , obtained in our previous researches and two references.
In Discussion part after we stated "Magnetic measurements support the idea of applying the as prepared dental magnetic material on the tooth surface under an external magnetic field, method with a major impact in reducing the occurrence of dental caries under filling materials as well as in reducing microfractures." we add the two references that we cited in the Introduction too.
“line 239: The method of separation of magnetic nanoparticles should be specified in more detail.”
Answer: At the end of section 3.1, where the magnetic separation process was mentioned for the first time, a text has been introduced describing in more detail what the magnetic separation process consists of.
“Magnetic separation is an easy and highly efficient separation process of separating components of mixtures by using a magnet to attract magnetic substances. In our particular case, the process was used for magnetic separation of magnetite nanoparticles by the reaction mixture, also in the washing process of particles. In a typical separation process, the MNPs colloidal solution was placed on a permanent magnet, left for 10 minutes to separate, by migrating MNPs on the magnet. The supernatant was subsequently discharged; the sample was thoroughly washed and subjected to a new magnetic separation process. At the end the clean MNPs was used in subsequent reaction steps, being redispersed in toluene.”
In the subsequent sections where magnetic separation is mentioned, we have added in brackets “as described previously”
“3. line 336: "the saturation magnetization value still remaining high enough for the intended application." Which criterion was used to support this statement? What is the limiting magnetization acceptable for intended application?”
Answer: Considering that in our targeted application, the magnetic properties of dental composite material are strictly used for the accurate localization of filling material in dental cavities, the saturation magnetization value obtained for dental composite material Ms=11.4 emu/g is considered a relatively good value for this kind of composite material, magnetic particles coated with double layer of inorganic non-conductive material.
In the following, I provide some literature references where similar composites, either coated with inorganic layer, or with organic polymeric layer (so non-magnetic layers) and with similar saturation magnetization values, are successfully used in various biomedical applications.
- single layer SiO2 covered MNPs : Ms = 28 emu/g (Islam, md.Abbas, Mohamed & Sinha, Brajalal.Joeng, Jong-Ryul.Kim, Cheolgi. (2013). Silica Encapsulation of Sonochemically Synthesized Iron Oxide Nanoparticles. Electronic Materials Letters. 9. 817-820. 10.1007/s13391-013-6019-1.
- sulfonic functionalized single layer SiO2 covered MNPs : Ms = 10.43 [Fatemeh Alemi-Tameh1 • Javad Safaei-Ghomi1 • Mohammad Mahmoudi-Hashemi1 • Raheleh Teymuri, Res Chem Intermed (2016) 42:6391–6406 DOI 10.1007/s11164-016-2470-6 2]
- polymer based Fe3O4@SiO2 nanocomposites: Ms=0.88 emu/g, polymer based Fe3O4@TiO2nanocomposites: Ms=2.84 emu/g; [Barrera, Gabriele, Paola Tiberto, Paolo Allia, Barbara Bonelli, Serena Esposito, Antonello Marocco, Michele Pansini, and Yves Leterrier. 2019. "Magnetic Properties of Nanocomposites" Applied Sciences 9, no. 2: 212. https://doi.org/10.3390/app9020212]
- polymeric nanocomposite: Ms= 15 Am2/kg (emu/g) [Ziolo, R.F.; Giannelis, E.P.; Weinstein, B.A.; Ohoro, M.P.; Ganguly, B.N.; Mehrotra, V.; Russell, M.W.; Huffman, D.R. Matrix-mediated synthesis of nanocrystalline gamma-Fe2O3-a new optically transparent magnetic material. Science 1992, 257, 219-223]
- Fe2O3/polymer nanoparticles: Ms=15.3 Am2 /kg (emu/g) [Vollath, D.; Szabó, D.V. Synthesis and properties of nanocomposites. Adv. Eng. Mater. 2004, 6, 117-127].
- conducting polymer based magneto composites: Ms=3.88 Am2/ kg(emu/g) [Yang, C.; Li, H.; Xiong, D.; Cao, Z. Hollow polyaniline/Fe3O4 microsphere composites: Preparation, characterization, and applications in microwave absorption. React. Funct. Polym. 2009, 69, 137-144.]

Reviewer 2 Report
The exciting subject related to utilising magnetic composite nanoparticles in dental structure restoration applications investigated by authors. The manuscript needs some improvements before the decision about readiness to be published in the Nanomaterials journal.
- The number of up-to-date references needs to be improved (already around 30% of references are published after 2018).
- The introduction section needs to be rewritten as the research gap and novelty are unclear. The last four paragraphs of the introduction section are more suitable for the "Materials and Methods" section as it explains the materials but does not introduce any references. It seems it is a technical note rather than a research article.
- All figures need improvement: The quality of the pictures should improve. All labels and axis labels need to be readable.
- The conclusion section size is unacceptable. The explanation needs to be summarised, and the current format is like repeating information in the "results" section.
- Considering the atomic weight for C, O, and Si, please explain how accurate XPS results are.
- Please explain how the micro-fracture will improve and reduce. Do you have any experimental evidence? If not, why is it part of your results and even introduced in your "abstract" section?
Author Response
Dear Reviewer
Please find enclosed our revised manuscript, "Novel magnetic composite materials for dental structure restoration application"
Authors: Izabell Craciunescu *, George Marian Ispas, Alexandra Ciorita, Cristian LeoÈ™tean, Erzsébet Illés, Rodica Paula Turcu
Manuscript ID: nanomaterials-2263544
The authors appreciate the efforts of the reviewer, being also grateful for their remarks and constructive suggestions helping us to improve the clarity of the manuscript. Following their observations, we have done efforts to improve the English and the clarity of the manuscript, doing our best to remove possible misunderstandings. The main amendments are highlighted in red in the present manuscript and a point-by-point response to the reviewer’ comments are presented in the following (answers in italics):
The exciting subject related to utilizing magnetic composite nanoparticles in dental structure restoration applications investigated by authors. The manuscript needs some improvements before the decision about readiness to be published in the Nanomaterials journal.
“The number of up-to-date references needs to be improved (already around 30% of references are published after 2018)”
Answer: To improve this aspect, a total of 29 new references of which 15 are post-2018 citations have been introduced in the article.
“The introduction section needs to be rewritten as the research gap and novelty are unclear. The last four paragraphs of the introduction section are more suitable for the "Materials and Methods" section as it explains the materials but does not introduce any references. It seems it is a technical note rather than a research article”
Answer: The Introduction section has been restructured by adding new paragraphs and highlighting the main purpose of the article and the novelty degree.
“All figures need improvement: The quality of the pictures should improve. All labels and axis labels need to be readable”
Answer: We improved the quality of the images as suggested.
“The conclusion section size is unacceptable. The explanation needs to be summarized, and the current format is like repeating information in the "results" section”
Answer: As you suggested, we re-evaluated the conclusions section.
“Considering the atomic weight for C, O, and Si, please explain how accurate XPS results are.”
Answer: The determination of atomic concentrations from XPS has typically an accuracy of 10%.
“Please explain how the micro-fracture will improve and reduce. Do you have any experimental evidence? If not, why is it part of your results and even introduced in your "abstract" section?”
Answer: In our preliminary articles, reported as references [38] and additional one [46 ]
[38]. C Zaharia, C.; Duma, V.F.; Sinescu, C.; Socoliuc, V.; Craciunescu, I.; Turcu, R.P.; Marin, C.N.; Tudor, A.; Rominu, M.; Negrutiu, M.L. Dental Adhesive Interfaces Reinforced with Magnetic Nanoparticles: Evaluation and Modeling with Micro-CT versus Optical Microscopy. Materials, 2020, 13,3908.
[46]. C.Zaharia, C. Sinescu, A.G.Gabor, V.Socoliuc, S. Talpos, T. Hajaj, P. Sfirloaga, R. Oancea, M. Miclau, M.L.Negrutiu, Influences of Polymeric Magnetic Encapsulated Nanoparticles on the Adhesive Layer for Composite Materials Used for Class I Dental Fillings, MATERIALE PLASTICE, 56, 2, 2019 ]
there are already research proving the usefulness of magnetic nanoparticles in dental restoration applications. The conclusion of this studies demonstrate that the use of magnetic nanoparticles after incorporation into dental adhesives, can reduce the thickness of the adhesive layer by 30% by applying a magnetic field on the tooth surface for 2 min and by 86.5 % for applying the same magnetic field for 5 min compared to the application of dental adhesives by conventional techniques.

Round 2
Reviewer 1 Report
Authors properly completed their manuscript, defined the positive effect of magnetic nanoparticles in the dentistry, and added new arguments supporting their models. The manuscript may be published in Nanomaterials.
Reviewer 2 Report
Considering the changes implemented in the manuscript by the authors, it is now ready for publication.